# Annotating the Pandemic: Named Entity Recognition and Normalisation in COVID-19 Literature

**Nico Colic**[*]
colic@ifi.uzh.ch

**Lenz Furrer**[†]
furrer@cl.uzh.ch

**Fabio Rinaldi**[‡]
fabio@idsia.ch

## Abstract

The COVID-19 pandemic has been accompanied by such an explosive increase in media coverage and scientific publications that researchers find it difficult to keep up.

We are presenting a publicly available pipeline to perform named entity recognition and normalisation in parallel to help find relevant publications and to aid in downstream NLP tasks such as text summarisation. In our approach, we are using a dictionary-based system for its high recall in conjunction with two models based on BioBERT for their accuracy. Their outputs are combined according to different strategies depending on the entity type. In addition, we are using a manually crafted dictionary to increase performance for new concepts related to COVID-19.

We have previously evaluated our work on the CRAFT corpus, and make the output of our pipeline available on two visualisation platforms.

## 1 Introduction

The body of scientific literature is growing at an unprecedented rate, and this is particularly evident in the response of the biomedical research community to the 2020 COVID-19 pandemic. Several platforms have been established to track publications related to COVID-19, most prominently the COVID-19 Open Research Dataset (CORD-19)[1], a collaboration of the US Government and multiple other organisations, the LitCovid dataset, maintained by the NIH, which indexes papers published on PubMed related to the pandemic (Chen et al., 2020), or Novel Coronavirus Research Compendium (NCRC)[2], which contains 800 publications selected manually for their originality and quality.

In this publication, we are processing the articles of the LitCovid dataset, which at the time of writing contains almost 50 000 publications related the 2020 COVID-19 pandemic only, showing growth at a steady rate since its beginning.

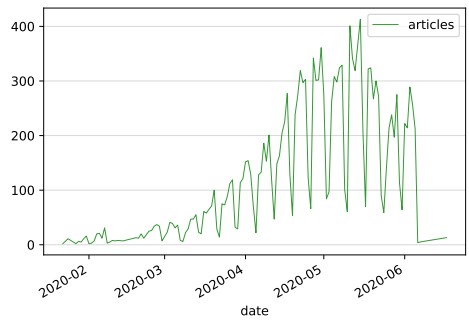

Figure 1: Publications per day included in LitCovid

The flurry of news and public discussions about the pandemic, which includes a substantial amount of fake news, has been termed *"infodemic"*. However, the term could be applied also to the rapid growth of reports and publications pertaining the disease (see Figure 1). Interestingly, this growth pattern seems to resemble that of the spread of the disease in western countries (with a delay of one to two months).

While the growth is not exponential as it has occasionally been reported, it is still far beyond what virologists and medical scientists can manually process. This is an exacerbation of a general problem in biomedical research, where researchers cannot keep up with the growth of literature that pertains to their research, and need to resort to named en-

---

[*] University of Zurich, Department of Computational Linguistics

[†] University of Zurich, Department of Computational Linguistics

[‡] Dalle Molle Institute for Artificial Intelligence Research (IDSIA); Swiss Institute of Bioinformatics; University of Zurich, Department of Computational Linguistics

[1] semanticscholar.org/cord19

[2] ncrc.jhsph.edu/

tity recognition (NER), named entity normalisation (NEN) and text summarisation technologies to identify relevant publications (Lu, 2011).

In NER, entities of interest are identified as text spans in free text; and then, in NEN, mapped to unique IDs in a controlled vocabulary. They constitute a fundamental step for other down-stream text processing tasks, on one hand; but are also a means to its own end, allowing publications to be indexed by the entities they contain, on the other hand.

In previous research, we have shown that we can obtain better results by performing NER and NEN in parallel rather than sequentially, avoiding propagation of errors between the steps. We are building on this previous research and add a further processing step to find terms specific to COVID-19.

## 2 Related Work

In March 2020, the US White House collaborated with the National Library of Medicine, the Allen Institue for Artificial Intelligence and other private companies to create the CORD-19 corpus (Wang et al., 2020a), and with it a set of 18 challenges such as *What do we know about COVID-19 risk factors?* for data scientists to participate in, hosted on Kaggle[3].

The response of the text mining community to the pandemic and such shared tasks has been enormous, producing a wide array of webservices, machine learning models and databases; usually adapting existing frameworks to suit the pandemic. Wang et al. (2020c), for example, are retraining SciSpacy on the CORD-19 corpus to improve its NER performance.

Some research has already been directed at downstream tasks, using a simple dictionary-based NER method as a base to perform entity relation extraction (Rao et al., 2020; Wang et al., 2020b), to create a knowledge base (Khan et al., 2020) or for summarisation systems (Gutierrez et al., 2020; Kieuvongngam et al., 2020).

The problem of NER and NEN in the biomedical domain, generally, has traditionally been approached with pipelines, using rules or dictionaries (Campos et al., 2013; D'Souza and Ng, 2015). More recently, however, machine learning using various architectures such as LSTMs or CRFs have become more popular (Leaman et al., 2013; Habibi et al., 2017).

---

[3]bit.ly/384VgBQ

In this vein, it has been suggested to approach NER and NEN simultaneously (ter Horst et al., 2017; Lou et al., 2017), which is similar to the approach that we follow.

The authors of the LitCovid data set, which we process in the present work, also perform NER and NEN on the dataset using PubTator (Wei et al., 2019). In their work, they annotate for 6 entity types (genes, diseases, chemicals, mutations, species and cells) and use a different architecture for every single type. For example, they use a linear classifier for annotating diseases (Leaman and Lu, 2016), and a BERT-based transformer for finding chemicals. This differs fundamentally from our approach, where we employ the same architecture for all entity types. Furthermore, apart from the NCBI Taxonomy, we are using different controlled vocabularies for entity normalisation for all types.

## 3 Pipeline

In our approach, we build on our previous efforts where we use a parallel architecture to perform NER and NEN simultaneously (Furrer et al., 2019a, 2020). Traditionally, NER and NEN are performed after each other, which means that spans of mentions of entities are identified first, and then mapped to the corresponding entry in a controlled vocabulary. This approach has the drawback that errors made in the first step are irrecoverably propagated to the second stage.

In our approach, however, we perform those two steps simultaneously, and were able to show that it outperforms the traditional approach (Furrer et al., 2019a). We are using BioBERT, a pre-trained language model, which we trained on the CRAFT corpus, a collection of nearly 100 full-text medical articles manually annotated for 10 different medical entity types. We have evaluated our approach using the CRAFT corpus, and obtained F1-scores between 0.74 and 0.92 depending on the entity type.

To improve our results on COVID-19 literature, we are adding an additional step of post-annotating our results using a manually crafted dictionary specific to COVID-19.

### 3.1 Vocabularies

The dataset is annotated for entities coming from 10 different ontologies as they are used in the CRAFT corpus, such as *Chemical Entities of Biological Interest* (**CHEBI**) or the *NCBI Taxonomy*

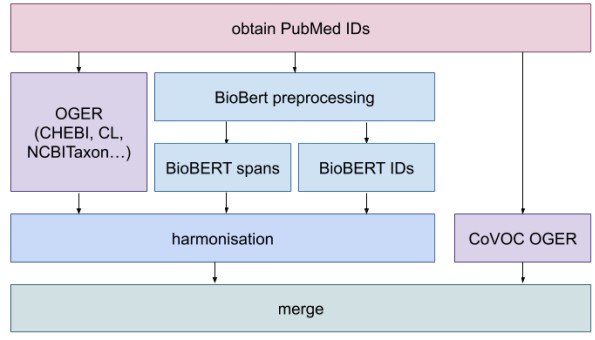

Figure 2: Overall structure of the pipeline

(**NCBITaxon**).

Additionaly, we employ a manually curated, COVID-19 specific terminology[4] containing over 250 terms. This is derived from the COVoc[5] vocabulary, developed by members of the Swiss Institute of Bioinformatics. We are using these ontologies because we were able to test our performance using the CRAFT corpus, and because they provide extensive coverage over the biomedical domain (Cohen et al., 2017).

## 3.2 OGER

OGER is a dictionary-based look-up tool using an efficient fuzzy matching algorithm (Furrer et al., 2019b). Relying on a dictionary mapping relevant entities to their ID, its performance depends on the dictionary's quality and extent, which manually or automatically curated ontologies such as CHEBI provide. It thus requires no training, and can detect entities that an example-based system would miss if they are not present in the training data, provided they are present in the dictionary.

## 3.3 BioBERT

BERT is a multi-layer transformer trained on the English Wikipedia and BookCorpus (Devlin et al., 2018). While it is trained to predict whether a sentence follows another and randomly blacked out words, the resulting language model can be fine-tuned for different tasks, such as NER (Hakala and Pyysalo, 2019) and NEN, or adapted for different domains through further training. BioBERT is the result of training BERT on PubMed articles, making it useful for biomedical applications (Lee et al., 2020; Sun and Yang, 2019).

We have used BioBERT and trained it further on the CRAFT corpus to build a span wprediction

---

[4] bit.ly/3jJxhgJ
[5] github.com/EBISPOT/covoc/

and an ID prediction model. The span predictor produces IOBES labels, and is used in conjunction with OGER to provide ID labels. The ID predictor also conceptualises NEN as a sequence tagging problem and works like a classical NER model, but with the output tagset extended to cover all possible concept labels.

The ID predictor thus predicts spans and IDs directly, making the use of other models theoretically superfluous. However, it suffers from the fact that it cannot predict concepts not seen during training and that it does not perform well for tokens that occur both in general domain language and in biomedical entities (such as *I* in *hexokinase I*). By using the span prediction model in conjunction with OGER, too, we alleviate these shortcomings.

## 3.4 Harmonising, annotating for COVID-19, merging

For conflicting or overlapping annotations between the BioBERT span and ID classifiers as well as OGER, we were able to show in our previous work that the optimal merging strategy depends on the entity type in question (Furrer et al., 2020). In this step, we take these findings into account when deciding which system's output to prioritise for the final output. If a span prediction is given preference, the ID label as produced by OGER as described in Section 3.3 is used.

In a last step, we run OGER again to produce an additional layer of annotations for terms specific to COVID-19 using the COVoc vocabulary. In this way we hope to be able to maintain the accuracy of our models for the established vocabularies, while allowing for rapid changes to be made to the set of entities specific to the pandemic without having to retrain the BioBERT modules.

The outputs are then merged for all entity types, and converted to various formats.

## 4 Results

So far, with our pipeline we have processed over 33 000 abstracts from PubMed and 7883 full-text articles from PMC, with a total amount of over 400 000 and 900 000 annotations, respectively (see Table 1).

With our pipeline, we are able to continuously process new articles that are added to the LitCovid dataset, and distribute our annotations in the following ways:

- PubAnnotation and EuroPMC

| vocabulary | PM abstracts | PMC articles |
|---|---|---|
| CoVoc | 165668 | 261287 |
| UBERON | 79899 | 204355 |
| NCBITaxon | 67278 | 147524 |
| GO_BP | 34510 | 84604 |
| CHEBI | 30720 | 99673 |
| PR | 12319 | 48471 |
| GO_CC | 7656 | 28738 |
| CL | 7332 | 28849 |
| SO | 6801 | 25017 |
| MOP | 449 | 2559 |
| GO_MF | 73 | 260 |
| **total** | **412 705** | **931 337** |

Table 1: Annotations per vocabulary for PubMed and PMC

- Our own webservice using BRAT
- Freely downloadable files

The OGER annotations can be obtained through an API[6]. The code to run the pipeline[7], its outputs[8] as well as the CRAFT-trained BioBERT models[9] are publicly available, and with some effort could be modified using OGER's format conversion to process other dataset such as CORD-19.

### 4.1 Online Repositories

PubAnnotation is an online repository for annotations on PubMed articles, (Kim et al., 2015, 2019), which also features the annotation visualisation engine **TextAE** (see Figure 3). Europe PMC is a repository of publications akin to PubMed, but also allows display of annotations (Consortium, 2015). We uploaded our annotations to both services.

### 4.2 BRAT

On our own infrastructure[10], we host an instance of BRAT, which visualises annotations in a similar fashion as PubAnnotation (Stenetorp et al., 2012).

### 4.3 Downloads

To further facilitate down-stream tasks, we provide our annotations in the most frequently used annotation formats[11]: `.txt`, CoNLL `.tsv` and BioC `.json`.

---

[6]`bit.ly/2Vrbekw`
[7]`github.com/Aequivinius/covid`
[8]`bit.ly/3eMylOq`
[9]`doi.org/10.5281/zenodo.3822363`
[10]`bit.ly/3eITn0o`
[11]`bit.ly/386BbuN`

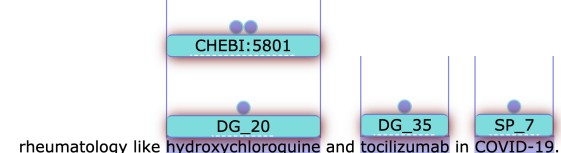
rheumatology like hydroxychloroquine and tocilizumab in COVID-19.

Figure 3: Annotations visualised by PubAnnotation's TextAE.

## 5 Evaluation

Given the recency of the pandemic, there is currently a lack of resources that allow evaluation of work on the COVID-19 literature. Without a gold standard we cannot offer a true evaluation. We hope to be able to test the efficacy of our own work in the future when such resources become available.

## 6 Discussion

Tools that automatically process literature related COVID-19 generally fall into two broad categories: Systems that follow some sort of text summarisation approach, and NER+NEN systems.

Much attention has been directed at previously mentioned Kaggle challenge, for which over 1500 solutions have been submitted, ranging from statistical data exploration to a full clustering of the literature. One of the top submissions[12], for example, attempts to identify risk factors of COVID-19 by applying unsupervised topic modeling algorithms. Such approaches are very common among the submissions, but suffer from a high number of false positives.

Similarly, platforms that allow browsing corpora of COVID-19 papers such as COVIDScholar[13] and the BERT-driven COVID-19 Research Explorer[14] rely on word embeddings and other unsupervised algorithms to find matching publications or even passages in publications. For the latter, the authors attempt to go beyond traditional document retrieval, and employ an automatically generated corpus to fuel their question answering learning (Ma et al., 2020). However, such approaches lack the precision typical NER+NEN-driven approaches offer, and don't perform particularly well at matching entity synonyms due to their representation as high-recall word vectors rather than precisely matched entities.

For example, both applications yield different

---

[12]`bit.ly/2VkN6QP`
[13]`covidscholar.org/`
[14]`bit.ly/3fWNOLG`

results for either *Angiotensin converting enzyme 2* or *ACE2*, even though the terms are equivalent (and link to the same entry in the Protein Ontology). Repositories that perform controlled vocabulary NEN such as KnetMiner, for example, avoid this error (Hassani-Pak et al., 2020).

Services exploring the scientific literature still fall in *either* of the two camps, and thus fail to exploit the high precision benefits NER+NEN offers and the variety of applications text summarisation approaches afford simultaneously.

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
