# OpenReview forum: "Annotating the Pandemic: Named Entity Recognition and Normalisation in COVID-19 Literature"
_EMNLP/2020/Workshop/NLP-COVID — NLP-COVID19-EMNLP Poster_

### Official Review · AnonReviewer2 · 2020-09-14
**Useful resource, but how much of the paper is novel?**

**Rating:** 6
**Confidence:** 4

**Review:**

The authors describe a pipeline for Named Entity Recognition and Normalisation on the Covid-19 literature, with the results being distributed to the public via various means - PubAnnotation, EuroPMC, their own BRAT webservice, and downloads in four formats. The code to run the pipeline and the trained models are also made available. The system continuously responds to updates in the LitCovid dataset. Much of their work uses a system that has previously been evaluated in a shared task - however they also add a Covid-19 specific terminology, COVoc.

Reasons to accept: This paper presents a useful resource to community, in a way that is ready for use. Most of the methods used have been previously evaluated, so the quality is known. COVoc is also a potentially useful resource.

Reasons to reject: Lots of the paper is spent describing the technical details of the parts of the system that have already been published and evaluated - space that would be better used to describe the novel contributions. For example the whole description of COVoc is "Additionaly, we employ the manually curated, COVID-19 specific terminology COVoc, containing over 250 terms" and a URL for obtaining COVoc - substantially more on COVoc would be welcome. Given this imbalance, it is hard to tell how much original work is being presented in this paper. Much of the system _has_ been evaluated as part of a shared task, but no other teams contributed to that facet of the shared task, making it hard to know how the system compares with the state of the art. The authors state that the pipeline "with some effort could be modified using OGER's format conversion to process other dataset such as CORD-19" - CORD-19 is an important resource and having ready-to-go annotations for this would be very useful.

Conclusion: Despite my misgivings about the level of novelty in this paper, I think the public annotations are potentially of use to the community, and having a citable publication associated with the annotations would expedite use of these annotations, in an area where time is of the essence.

---

### Official Review · AnonReviewer1 · 2020-09-16

**Rating:** 6
**Confidence:** 4

**Review:**

This study aimed to improve the CORD-19 dataset and provide documentation on how the public can access their pipeline. This study understated the problem and the methods used to improve the CORD-19 dataset do not appear novel.  While this study provides an open source that may be useful to the public and scientists, its technical contributions were unclear.

---

### Official Review · AnonReviewer3 · 2020-09-25
**Good work but needs additional baselines and analysis**

**Rating:** 5
**Confidence:** 4

**Review:**

This paper is built on their previous work on a parallel architecture to perform named entity recognition and normalisation simultaneously. They use a dictionary-based system in conjunction with two models based on BioBERT, whose outputs are combined as per the entity type. They also use a manually crafted dictionary optimized for the new COVID-19 concepts.

The paper overall is written well: easily understandable and transparent. Motivation is well established. Their willingness to share the code, outputs, BioBERT models and even the annotations is appreciable. However, the paper lacks novelty, rigor and doesn’t attempt to compare against other baselines. They should definitely try to add additional baselines in the event that the paper is accepted